# Neural mechanisms of emotion during shifting perspectives and recognizing new information: An fMRI study

Kazutaka Ueda, Xiaoxiang Wu, Hideyoshi Yanagisawa *

Department of Mechanical Engineering, Graduate School of Engineering, The University of Tokyo, Tokyo, Japan

* hide@mech.t.u-tokyo.ac.jp

## Abstract

In generating novel ideas during the creative process, what ignites positive emotions like "Aha" experiences of insight? This study explored this by validating the biological plausibility of a mathematical model predicting emotions when shifting perspectives to recognize alternate information regarding an event. Using functional magnetic resonance imaging, we assessed brain activity as participants watched card magic videos with experimentally varied ease of information recognition. The results indicated that when shifting from certain to uncertain belief-based recognition, subjective interest arises if the new recognition is distant from the prior certain belief but close to the subsequent uncertain belief, accompanied by brain activations related to positive emotions. These findings suggest that interest emerges when deviating from conventional ideas towards unexpected yet easily comprehensible new ones, providing strategic insights for ideation.

## Introduction

Creative processes (creativity) exist in all aspects of our lives, including science, technology, education, and culture, and serve to solve problems and enrich our lives [1]. When faced with a problem, we attempt to solve it by accessing our memories and information from the outside world [2]. We deductively formulate hypotheses based on our experiences and inductively test them to reach a solution [3]. However, complex or new problems that have not been experienced previously may not be solved easily. In such cases, we actively access our memories and information from the outside world, motivated by problem solving. By shifting perspectives, we recognize strange or surprising events and generate hypotheses through abduction [4]. In this regard, the process of "insight," in which sudden inspiration is obtained, is necessary. When an idea obtained through insight is starkly different from what was previously thought, positive emotions such as interest and surprise are generated [5]. Such an experience is called the "Aha" experience or the Eureka effect [6]. In the

**Data availability statement:** All relevant data are within the paper and its Supporting information files.

**Funding:** This study was supported by Japan Society for the Promotion of Science URL of the funder website: https://www.jsps.go.jp/english/e-grants/ in the form of a grant awarded to H.Y. (21H03528 and 25H01132). The specific roles of this author are articulated in the 'author contributions' section. The funders had no role in study design, data collection and analysis, decision to publish, or preparation of the manuscript.

**Competing interests:** The authors have declared that no competing interests exist.

scientific field, the positive emotion of an "Aha" experienced with a new discovery that overturns conventional wisdom increases the spirit of inquiry for further discovery [7,8]. In the field of technology, the "Aha" experience with a major shift in perspective upon developing a new technology triggers the next technological innovation. Regarding the process of insight, the fluency theory [9] explains that the reason for positive emotions during this process is that cognitive processing of information becomes smoother, leading to a heightened sense of confidence when new information is recognized, namely imbuing a sensation of being certainly convinced. The pleasure interest model of aesthetic liking (PIA model) [10] divides human cognitive processing into unconscious automatic and conscious controlled processing. When people can recognize information smoothly during automatic processing, they have high fluency and experience pleasant feelings of *pleasure*; however, when they cannot do so, they have low fluency and experience unpleasant feelings of *displeasure*. The emotional response is caused by the disfluency reduction to the information, i.e., the resolution of the uncertainty of recognition, which is a process of control to understand the information further. When disfluency reduction occurs for information, positive feelings of *interest* are also generated for other information. Contrarily, when disfluency reduction does not occur, the information cannot be recognized, and a negative emotion called *confusion* is generated. In the insight process, the positive emotion of *interest* is considered the positive emotion of the "Aha" experience when other information is recognized by changing the perspective.

## Prediction about the emergence of interest from the free-energy model

Our group proposed the general framework of a mathematical free-energy model of emotional valence to explain emotions such as *pleasure* with recognition of information and *interest* with shifting perspectives and recognition of other information [11]. Based on the minimized free-energy model of emotion [12] and the PIA model [10], we formulated the change in the fluency of information processing in unconsciously automatic and consciously controlled processes when recognizing information from one perspective and from another perspective as an increase or decrease in free energy.

Consider an instance of observing the hidden Dalmatian dog illusion. At first glance, a cluster of black dots is perceived. This initial perception is an unconscious, automatic process that results in *pleasure* due to the recognition of a familiar pattern. This process is characterized by high information processing fluency and decrease in free energy ($\Delta F_i$ in Fig 1). If this image is attempted to be viewed from another perspective, the process shifts to conscious and controlled, leading to reduced fluency in information processing and increased free energy. If one cannot make sense of another perspective, one might feel negative emotions, such as confusion. However, upon deducing that the image represents a Dalmatian dog, the inefficiency in processing the new pattern is resolved. This resolution, known as the "disfluency reduction," decreases the free energy again and leads to an "Aha" experience, which is accompanied by a positive emotional response. We interpret this emotion as that of interest.

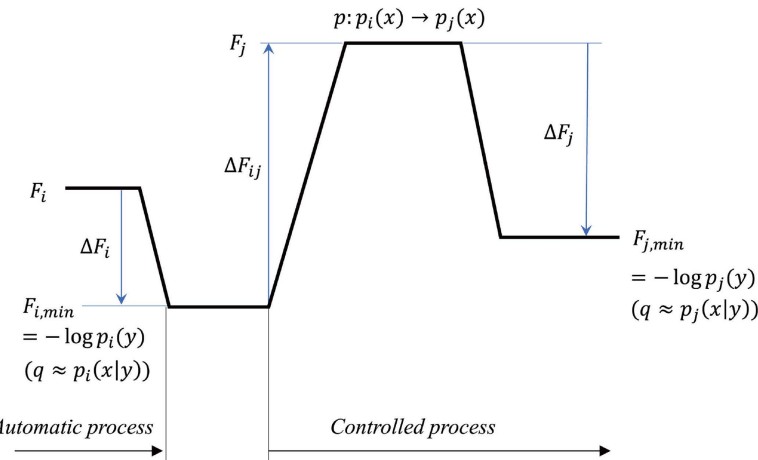

**Fig 1. Free-energy variations in the dual process and emotions.** A model illustrating the formulation of changes in the fluency of information processing within unconscious automatic and consciously controlled processes when recognizing information from one perspective versus another, represented as increases or decreases in free energy.

In our model, we quantified the amount of reduction in free energy ($\Delta F_j$ in Fig 1) upon the recognition of something from another perspective. We calculated this reduction based on the following factors: how certain we were about what we first recognized ($s_i$ in Fig 2, i.e., prior belief uncertainty of the first recognized information: the variance of the prior distribution [$p_i(x)$ in Fig 2, i.e., a prior belief of first recognized information]), how certain we are about another recognition ($s_j$ in Fig 2, i.e., prior belief uncertainty of alternative recognized information: the variance of the prior distribution [$p_j(x)$ in Fig 2, i.e., the prior belief of alternative recognized information]), how different the prior belief of another recognition is from the first one and recognized information later, i.e., the recognition distance.

The recognition distance is the ratio of two distances, where the first distance is between the recognized information later ($p(y|x)$ in Fig 2) and the prior belief of alternative recognized information ($\delta_j$ in Fig 2). The second distance is between the prior distribution ($\mu_{ij}$ in Fig 2).

We used numerical simulations to understand how the recognition distance affected the amount of free-energy reduction. In other words, we studied how expanding our understanding of information (disfluency reduction) depended on how much we ought to change our perspective.

We predicted that disfluency reduction would be greater for larger recognition distances when the belief uncertainty for the first recognized piece of information ($s_i$) was larger than that for another recognized piece of information ($s_j$). Therefore, in the hidden Dalmatian dog illusion, when our prior beliefs are highly uncertain, such as the set of black dots that we initially recognized (i.e., the arrangement of the dots is random), and when our prior beliefs are more certain, such as another recognized Dalmatian dog (i.e., we are familiar with the features of the Dalmatian dog), a larger recognition distance (i.e., we recognize the Dalmatian dog only from the observed picture) causes more surprise and interest once the Dalmatian dog is recognized.

However, this mathematical model predicts that the amount of disfluency reduction is larger when the uncertainty of belief in alternative recognized information ($s_j$) is larger than the uncertainty of belief in the first recognized information ($s_i$), when the recognition distance is smaller. This means that when recognition switches from information with certain beliefs to information with uncertain beliefs, the change in recognition is easier and more interesting. The predictions of this mathematical model explain the shift from existing solid knowledge (i.e., common sense) to unprecedented and uncertain knowledge (i.e., ideas that nobody has thought of before) in the creative process of science and technology. The model

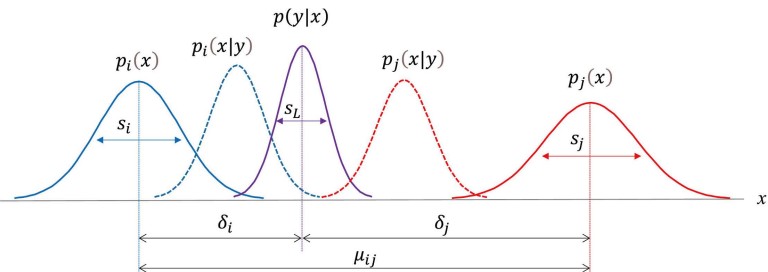

**Fig 2. Gaussian form of Bayesian priors and posteriors in the dual process.** Perspective *i* is recognized in the automatic process with prior $p_i$, followed by perspective *j* being recognized in the controlled process with prior $p_j$ from the same observation *y*. The likelihood of the observation is assumed to be placed between the two perspectives. The priors' mean distance $\mu_{ij}$ is a summation of the prediction errors $\delta_i$ and $\delta_j$.

predicts that, in this case, the easier it is to change recognitions (i.e., the less surprising another idea is), the more positive feelings of interest arise. The fact that less unexpectedness is more interesting differs from our intuition and is based on our experience. However, if we look at this situation from a different perspective, the recognition distance is proportional to the distance between the observed information and the prior distribution of the newly recognized information. Furthermore, a small recognition distance indicates that another idea is far from the existing solid knowledge. In other words, an idea may be interesting if nobody has ever thought of it before and if it is easy to recognize and accept as another idea. Therefore, the predictions of our model are important for understanding processes that are essential for creativity, such as abduction.

This study aims to demonstrate the biological validity of the predictions of a mathematical model that explains the emotions involved in shifting perspectives and recognizing alternative information. Specifically, by measuring human brain functions, we tested the prediction that a smaller recognition distance (i.e., easier recognition) would produce more positive emotions, such as interest, when shifting from recognition based on certain beliefs to recognition based on uncertain beliefs.

## Neuroscientific research related to interest

A functional brain measurement study investigating the transition of emotion and brain processing in the automatic and deliberate process showed that during preference judgments for human faces, rapid automatic processing occurs in the nucleus accumbens, followed by slower control processing in the orbitofrontal cortex (OFC) and insula [13]. Additionally, neuroimaging studies of positive emotion [14–17] have shown that the anterior cingulate cortex [18–20], amygdala [21–25], hippocampus, striatum, caudate, and putamen are involved in processing positive emotions. Furthermore, hippocampal and parahippocampal activity during the presentation of an interesting question [26] is activated during positive emotions, which facilitates subsequent memory recall [27], suggesting that memory retrieval and consolidation are facilitated during positive emotions such as interest.

The purpose of this study is to analyze the brain activity related to subjective interest and positive emotions associated with changes in the recognition of information, as well as to verify the validity of model predictions based on the free-energy model of emotion. We experimentally created a situation wherein participants shifted from certain recognition to uncertain recognition by watching card magic, namely a trick where the objective is to determine the position of the target card, and experimentally manipulated the size of the recognition distance by the difficulty of recognizing the card magic technique (the trick). Brain activity during the task was measured using functional magnetic resonance imaging (fMRI), and the aforementioned brain regions were analyzed as the regions of interest (ROIs).

## Materials and methods

### Participants

In total, 20 healthy participants (10 males, 10 females, aged 24.3 ± 3.3 years, all right-handed) who had not mastered card magic skills were included in this study. This study was approved by the Research Ethics Committee of the Graduate School of Engineering at the University of Tokyo (approval number: KE22−3). Written informed consent was obtained from all the participants. All procedures conducted in this study complied with the guidelines and regulations established by the Research Ethics Committee of The Graduate School of Engineering at the University of Tokyo.

### Stimuli, task, and design

In this experiment, the participants were requested to estimate the position of the target card by watching a video of card magic. Twelve card magic techniques were used in this study. The participants watched three videos of each technique and estimated the position of the target card based on the information provided by the video content. In the first video (S1 Video), a specific card was indicated by an arrow as the target card. The second (S1 and S2 Videos) and third (S3 Video) videos were designed to change participants' perceptions of the target card's position.

We manipulated the uncertainty of beliefs regarding the recognized information in our mathematical model by the uncertainty of the position of the target card in the card magic trick. We also manipulated the recognition distance by changing the ease of estimating the true position of the target card. Furthermore, we designed the videos in such a way that the ease of estimating the card's true location varied depending on the second video (S2 Video). The condition in which it was easy to estimate the true position of the card was defined as the low-difficulty condition (S2 Video), while the condition in which it was difficult to estimate the true position of the card was defined as the high-difficulty condition (rewatch S1 Video).

The length of all videos was set to 14 seconds, and they were edited so that the participants' recognition would change 9 seconds after the beginning of S3 Video.

All participants encountered high- and low-difficulty conditions (within-participant factorial design). The order of the high- and low-difficulty conditions was counterbalanced across participants (refer to Stimulus.xlsx in the S1 File).

The following is a description of the trial flow using one technique of playing card magic as an example:

1. From the spectator's point of view (S1 Video, S1 Fig). First, the magician held a deck of cards facing down in his right hand. Subsequently, the cards were individually dropped onto the left hand. This technique is known as dribbling. The dribble stopped midway, and a card was selected and shown to the camera. The card indicated by an arrow was the target card (five of spades). The target card was then returned to its original position, and the dribbling continued. The magic trick ended when the top card dropped onto the left hand. This video was designed to make the participants guess that the final position of the target card was in the middle of the deck of cards. In other words, the final position of the target card in S1 Video was easy to understand, with low uncertainty.

2. For the next step, in the high-difficulty condition, participants rewatched S1 Video. However, in the low-difficulty condition, participants watched the same technique as in S1 Video, but from a different audience perspective. S2 Video for the low-difficulty condition was created so that the final position of the target card could be easily confirmed (S2 Fig). Specifically, the target card was selected from the middle of a deck of face-up cards and displayed to the camera. Next, the target card was turned over and returned to its original position. Only the target card was turned face down. Subsequently, as shown in S1 Video, the cards held in the right hand were dropped individually into the left hand using a dribble. At the end of the video, a face-down card was placed on top of the deck. This was the target card, which was supposed to be in the middle of the deck; however, it was moved to the topmost position of the deck.

3. S3 Video was filmed from the magician's point of view (S3 Fig): In S3 Video, a card held in the right hand was dropped individually into the left hand using a card dribble. Next, the magician stopped midway and selected a card (the target card) to be shown to the camera. The player then returned the target card to its original position and continued the dribbling. The technique was the same as that in S1 Video, except from a different perspective. In S3 Video, a clue to the technique of the card magic began 9 seconds after the start of the video. In the example shown in S3 Fig, in S3 Video, just before continuing to dribble, the little finger pulled the target card to a vertical position. S3 Video ended at this stage, so that the final position of the target card was unknown (forming an uncertain belief).

## fMRI scanning protocol

For fMRI, we used the 3T Magneton Prisma (Siemens Medical Systems, Erlangen, Germany) with gradient echo and echo planar imaging (EPI) sequences. The interval between two successive acquisitions of the same image (TR) was 3,000 ms and the echo time (TE) was 30 ms. Furthermore, volumes were acquired with 96 interleaved slices of 2.0 × 2.0 × 1.5 mm voxel size.

In contrast, for anatomical MRI, we used a T1-weighted gradient-echo pulse sequence (TR = 2,300 ms, TE = 2.26 ms; field of view = 256 mm; voxel dimensions = 1 × 1 × 1 mm) that facilitated localization.

## fMRI data preprocessing and analysis

Brain activity in the whole brain and ROIs was analyzed using the Statistical Parametric Mapping (SPM 12) software (Wellcome Department of Cognitive Neurology, London, UK). Additionally, functional connectivity between the ROI was analyzed using the SPM toolbox CONN. These analyses were performed on MRI signals from 9 to 15 seconds after the start of S3 Video of the low- and high-difficulty conditions. In the brain activity analysis, all the EPI images were spatially normalized using the Montreal Neurological Institute T1 template for group analysis. Moreover, imaging data were corrected for motion and smoothed using an 8-mm full-width half-maximum Gaussian filter. A general linear model (GLM) was employed to conduct first-level analysis. The GLM incorporated two regressors, each representing the low- and high-difficulty conditions. The regressors for both the low- and high-difficulty conditions had an onset of 9 seconds after the start of S3 Video and a duration of 6 seconds. Using second-level analysis based on a random-effects model, we identified the regions that showed significant responses during the task. Based on the neuroscientific research on interest cited in the Introduction, we selected the following brain regions as ROIs due to their established involvement in interest processing: the anterior cingulate cortex, amygdala, insula, OFC, parahippocampus, hippocampus, caudate, and putamen. In the analysis of brain activity, all structural ROIs were obtained from the WFU PickAtlas within SPM 12. In the whole-brain analysis, we used family-wise error correction at a threshold of $p < 0.05$ to control for multiple comparisons. In the ROI analysis, we opted for a $p$ threshold of 0.005 without applying a correction or establishing a cluster-level criterion. In the functional connectivity analysis using CONN, the ROIs were further subdivided into the following brain regions as nodes: the anterior cingulate cortex (AC), right and left amygdala (amygdala r, amygdala l), right and left insula (IC r, IC l), right and left OFC (FOrb r, FOrb l), right and left anterior parahippocampus (aPaHC r, aPaHC l), right and left posterior parahippocampus (pPaHC r, pPaHC l), right and left hippocampus (Hippocampus r, Hippocampus l), right and left caudate (Caudate r, Caudate l), and right and left putamen (Putamen r, Putamen l). In the first-level analysis, correlation coefficients were computed for all connections between nodes. The second-level analysis then tested these correlations for statistical significance using a cluster-level false discovery rate corrected threshold of $p < 0.05$.

## Results

### Subjective evaluation

Fig 3 shows a scatter plot of individual subjective scores for comprehending the card magic technique, with the high-difficulty condition on the horizontal axis and low-difficulty condition on the vertical axis. The subjective scores for the

**Fig 3. Understanding of card magic techniques (N = 20).** Scatter plot comparing subjective scores for comprehending the card magic techniques between high-difficulty (horizontal axis) and low-difficulty (vertical axis) conditions, illustrating higher recognition ease in the low-difficulty condition. Each dot represents an individual participant, and the number indicates the participant ID.

low-difficulty condition were higher than those for the high-difficulty condition, indicating that the technique was more easily recognized in the low-difficulty condition.

Further scatter plots of individual subjective scores for the degree of admiration and interest of the technique are shown in Figs 4 and 5, respectively (data input errors in the degree of admiration for Participants 1 and 2 were excluded from the figures). Moreover, Fig 5 shows that the five participants' subjective interest in the technique was higher in the high-difficulty condition; however, as a whole, subjective interest was higher in the low-difficulty condition.

The mean subjective scores for all the participants are shown in Fig 6. For each subjective score, a two-tailed paired t-test was conducted with the condition as the independent variable and score as the dependent variable. We found significant differences in comprehension scores ($t(19)=5.90$, $p=0.00$, Cohen's $d=1.05$, 95% CI $=0.83–1.74$), degree of admiration ($t(17)=3.63$, $p=0.00$, Cohen's $d=0.72$, 95% CI $=0.36–1.23$), and interest ($t(19)=2.27$, $p=0.03$, Cohen's $d=0.49$, 95% CI $=0.05–1.15$). Furthermore, both subjective scores were significantly higher under the low-difficulty condition than under the high-difficulty condition.

**Brain activity linked to interest**

S3 Video presents the magician's point of view in both the conditions, and it provides a clue about the technique of the card magic 9 seconds after the video begins. Therefore, we expected that the participants understood the technique and started to change their perception after 9 seconds of watching the video, and that between 9 and 15 seconds of watching the video, the low-fluency condition would be resolved, and a positive emotion, interest, would occur. We compared the

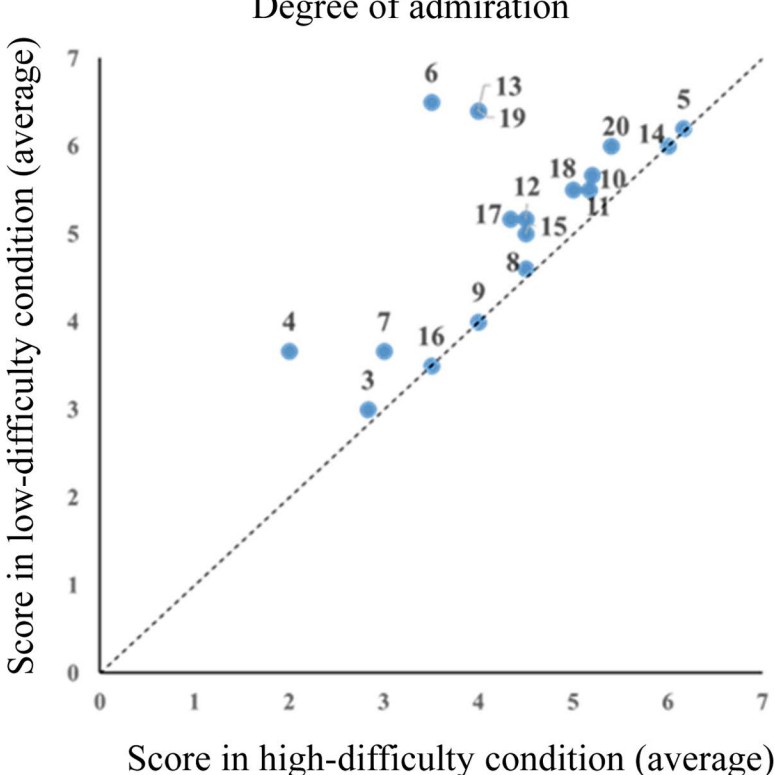

**Fig 4. Degree of admiration for card magic techniques (N = 18).** Scatter plot of individual subjective scores for the degree of admiration of the techniques, indicating a higher level of admiration in the low-difficulty condition. Each dot represents an individual participant, and the number indicates the participant ID.

MRI signals of the whole brain during the period between 9 and 15 seconds from the start of the video in the low- and high-difficulty conditions, but no significant differences were observed. In contrast, the ROI analysis revealed that activity in the anterior cingulate gyrus and hippocampus was greater in the low-difficulty condition than that in the high-difficulty condition (Fig 7). To examine how each brain region in the ROI worked together between 9 and 15 seconds in S3 Video, functional connectivity analysis was performed for each condition (Fig 8). In the high-difficulty condition, strong connectivity was observed in the caudate, putamen, insula, and OFC. For the low-difficulty condition, besides the connectivity between the aforementioned regions, there was strong connectivity between the left amygdala and hippocampus as well as between the right amygdala and hippocampus.

## Discussion

In this study, we developed a mathematical model to explain the emotions involved in shifting perspectives and recognizing alternative information [9]. The validity of this prediction was demonstrated by measuring fMRI while the participants watched a card magic video wherein the ease of recognition was experimentally manipulated.

The results showed that the degrees of subjective admiration and interest were higher in the low-difficulty condition, wherein the true position of the target card was easier to estimate, as compared to that in the high-difficulty condition, wherein the position of the target card was harder to estimate. This was tested when the participants first watched a card magic video (S1 Video) that they had a certain belief about, and they subsequently watched S3 Video, for which they had

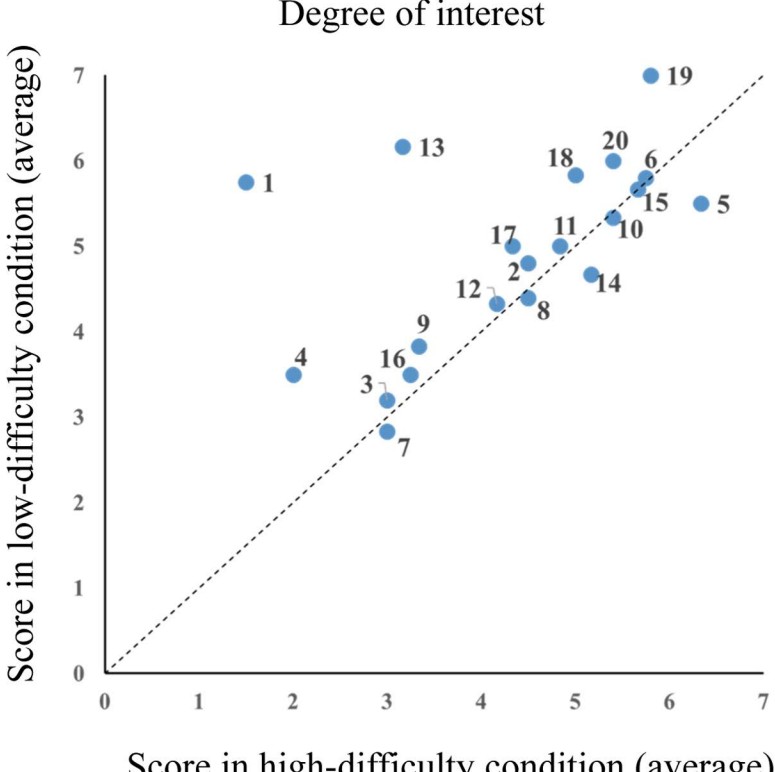

**Degree of interest**

**Fig 5. Degree of interest in card magic techniques (N = 20).** Scatter plot of individual subjective scores for the degree of interest in the techniques, showing moderately higher interest in the high-difficulty condition among five participants, though overall interest was greater in the low-difficulty condition. Each dot represents an individual participant, and the number indicates the participant ID.

an uncertain belief. This was consistent with the model prediction that upon shifting from recognition based on certain beliefs to recognition based on uncertain beliefs, more positive emotions such as interest are generated with a smaller recognition distance (i.e., easier recognition).

In addition, the low-difficulty condition showed greater activity in the anterior cingulate cortex and hippocampus as compared to the high-difficulty condition while watching S3 Video, wherein the information was recognized from different perspectives. These results suggest that when recognition shifts from a certain belief to a more uncertain one, greater activity in the cingulate gyrus [18–20] and hippocampus [26] is observed, particularly in cases in which the recognition distance is relatively small and recognition becomes easier. Given that both regions have been previously associated with the processing of positive emotional experiences, this neural pattern supports our model's prediction that recognition distance, rather than belief uncertainty alone, is a key determinant of disfluency reduction and emotionally engaging responses. Additionally, in the low-difficulty condition, increased functional connectivity between the amygdala [21–25] and hippocampus was observed. The amygdala is involved in processing emotionally salient stimuli, regardless of valence. This result may reflect heightened emotional salience during successful perspective shifts. This interpretation aligns with the model's assumption that when perspective is shifted from a certain belief to a more uncertain belief, and the recognition distance is relatively small, greater disfluency reduction and emotional engagement are more likely to occur.

Although our interpretation emphasizes that increased activation and connectivity in regions such as the anterior cingulate cortex, hippocampus, and amygdala may reflect disfluency reduction and emotional engagement during successful

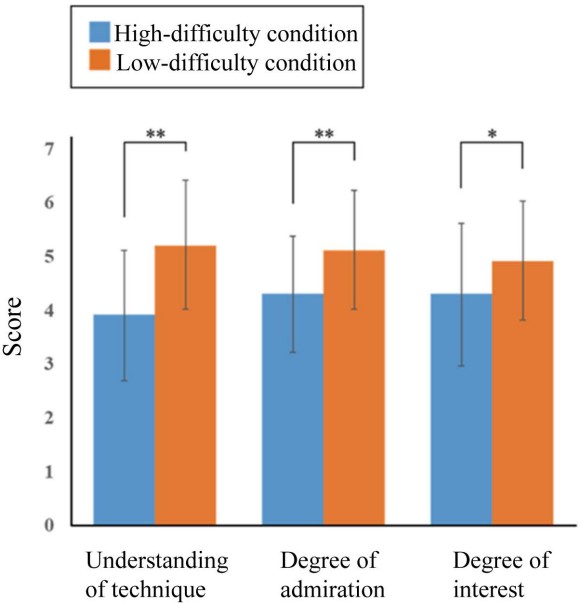

**Fig 6. Mean and standard deviation of each subjective score.** Mean subjective scores for comprehension, admiration, and interest across participants with two-tailed paired t-test results (**$p < 0.01$, *$p < 0.05$). Across all measures, subjective scores were significantly higher in the low-difficulty condition compared to those in the high-difficulty condition.

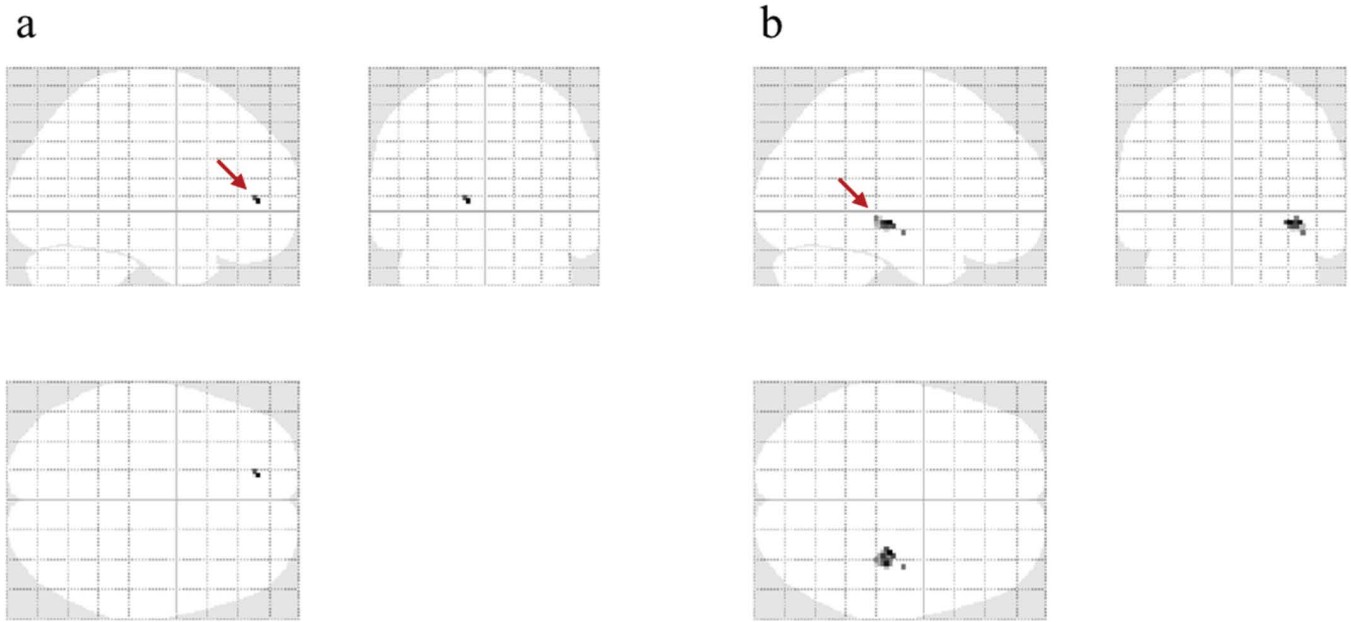

**Fig 7. Brain activity in the task (a: anterior cingulate cortex, b: hippocampus).** MRI signal comparison of the ROI in the anterior cingulate gyrus (red arrow in panel a) and hippocampus (red arrow in panel b) between 9 and 15 seconds after video onset, showing greater activity in the low-difficulty condition compared to that in the high-difficulty condition.

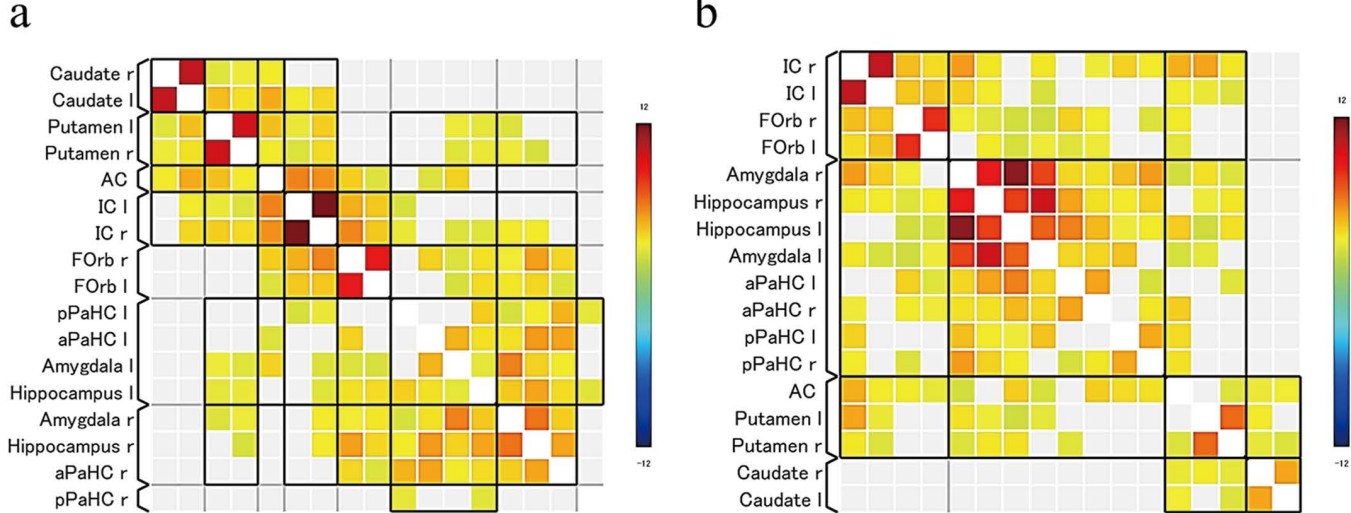

**Fig 8. Functional connectivity in the task (a: high difficulty condition, b: low difficulty condition).** Functional connectivity analysis of ROI between 9 and 15 seconds in both conditions, highlighting strong connections in the caudate, putamen, insula, and orbitofrontal cortex in the high-difficulty condition, and additional robust connectivity between the left and right amygdala and hippocampus in the low-difficulty condition.

perspective shifts, we acknowledge that other interpretations are also plausible. For example, anterior cingulate cortex activity may reflect general cognitive control or conflict monitoring [28] in response to updating beliefs. Hippocampal activity may reflect episodic memory retrieval [27] or novelty processing [29]. Similarly, increased connectivity between the amygdala and hippocampus in the low-difficulty condition may reflect enhanced attentional allocation to emotionally meaningful content [30]. These possibilities suggest that the observed neural responses may involve a combination of emotional, mnemonic, and attentional mechanisms. Future studies could incorporate more targeted designs to disentangle these processes.

A limitation of this experiment is that we did not collect information about participants' prior experience with magic tricks, their occupations in creative fields, or their general creative background. Although all participants had not mastered card magic skills, individual differences in creative training, profession, or familiarity with similar illusions could influence the ease of recognition or emotional responses. Future studies should consider assessing such background factors to better account for variance in emotional and cognitive responses.

The psychological phenomena investigated in our model predictions and human brain function measurement experiments—that is, the emotional reactions accompanying changes in the recognition of information—are thought to strongly influence creative behavior.

Among the rewards that determine behavior, internally generated desires and emotions may themselves be rewards, and the state in which we are motivated by these rewards is called intrinsic motivation [31]. Compared to extrinsic motivation, which is motivated by externally given rewards such as money or status, intrinsic motivation is more likely to be driven by satisfaction with work [32], motivation to learn [33], and well-being [34]. The emotion of interest is one of the determinants of intrinsic motivation [35]. We are intrinsically motivated by events and actions that we are interested in, while intrinsic motivation is suppressed for events and actions that we find boring [31]. In other words, the generation of positive emotions, such as interest, is considered to be an important condition for promoting actions to acquire information [32]. For example, the "Aha" experience during insight, accompanied by surprise, facilitates information processing and generates positive emotions, thereby promoting subsequent exploratory behavior [33]. In addition, when interest is aroused by the acquisition of alternative information, the hippocampus [26], which processes memories, is activated, and memory consolidation and recall are

promoted, which may encourage the next action. This finding could have a significant impact in two areas, i.e., creativity and education. Interest in uncertain perspectives and ideas is part of creative thinking: exploring new possibilities and combining seemingly unrelated information to generate new ideas. Abduction, i.e., hypothesis generation and inference, is an important part of a deeper understanding of this knowledge. Increasing interest in uncertain perspectives and ideas is deeply related to the abductive thinking process of generating new hypotheses and speculations. Abduction is inferring plausible explanations or solutions for unknown phenomena or problems. It is often used to find patterns in observational data and generate new theories or hypotheses to explain those patterns. Based on the findings of this study, when presented with an uncertain perspective or idea that deviates from common sense ideas and is unexpected, people attempt to generate a new explanation (hypothesis) to make sense of it and connect it to their existing perceptions. Furthermore, when this is supported by evidence close to cognitive distance, indicating that the conceived idea is not far-fetched, but rather feasible, interest in the viewpoint or idea is likely to increase. On the other hand, in education, this knowledge may improve the approach that learners use to acquire new knowledge and skills. Specifically, understanding new perspectives and knowledge that are distant from the learners' current beliefs and knowledge by using evidence that has a closer cognitive distance (concrete examples and experiences from daily life) may attract the learners' interest and deepen their understanding.

Based on the findings of this study, in the context of learning in education, it is necessary to discuss the effects of emotional reactions when shifting perspectives and recognizing alternative information. This includes the promotion of memory retention with the "Aha" experience by shifting perspectives and the promotion of intrinsic motivation (motivation to learn) for subsequent learning behaviors. Additionally, measures to increase job satisfaction and well-being are required and can be explored through these findings.

## Supporting information

**S1 Fig. Example of the flow of Supplementary Video 1 of the card magic trick (spectator's point of view).** First, the magician held a deck of cards facing down in his right hand. Subsequently, the cards were individually dropped onto the left hand. This technique is known as dribbling (leftmost panel). The dribble stopped midway, and a card was selected and shown to the camera. The card indicated by an arrow was the target card (five of spades) (second panel from the left). The target card was then returned to its original position, and the dribbling continued (third panel from the left). The magic trick ended when the top card dropped onto the left hand (rightmost panel).
(TIF)

**S2 Fig. Example of the flow of Supplementary Video 2 of the card magic trick (spectator's point of view where the position of the card can be easily checked).** In the high-difficulty condition, participants rewatched Supplementary Video 1. However, in the low-difficulty condition, participants watched the same technique as in Supplementary Video 1, but from a different audience perspective. Supplementary Video 2 for the low-difficulty condition was created so that the final position of the target card could be easily confirmed. Specifically, the target card was selected from the middle of a deck of face-up cards and displayed to the camera (leftmost panel). Next, the target card was turned over and returned to its original position. Only the target card was turned face down (second panel from the left). Subsequently, as shown in Supplementary Video 1, the cards held in the right hand were dropped individually into the left hand using a dribble (third panel from the left). At the end of the video, a face-down card was placed on top of the deck (rightmost panel). This was the target card, which was supposed to be in the middle of the deck; however, it was moved to the topmost position of the deck.
(TIF)

**S3 Fig. Example of the flow of Supplementary Video 3 of the card magic trick (magician's point of view).** In Supplementary Video 3, a card held in the right hand was dropped individually into the left hand using a card dribble (leftmost

panel). Next, the magician stopped midway and selected a card (the target card) to be shown to the camera (second panel from the left). The player then returned the target card to its original position and continued the dribbling (third panel from the left). The technique was the same as that in Supplementary Video 1, except from a different perspective. Just before continuing to dribble, the little finger pulled the target card to a vertical position. Supplementary Video 3 ended at this stage (rightmost panel).
(TIF)

**S1 File. Stimulus.xlsx: The order between the high-difficulty and low-difficulty conditions was counterbalanced among the participants, with 10 participants for each order.** The file "stimulus.xlsx" lists the video filenames used for each condition. For instance, for participants (N = 10) who underwent the high-difficulty condition first followed by the low-difficulty condition, they viewed the videos in the sequence of v1_1.avi, v1_1.avi, v3_1.avi for the high-difficulty condition, followed by v1_2.avi, v1_2.avi, v3_2.avi, continuing up to v1_6.avi, v1_6.avi, v3_6.avi (a total of 6 trials). For the subsequent low-difficulty condition, they viewed videos in the sequence of v1_7.avi, v2_7.avi, v3_7.avi, followed by v1_8.avi, v2_8.avi, v3_8.avi, continuing up to v1_12.avi, v2_12.avi, v3_12.avi (a total of 6 trials).
(XLSX)

**S1 Video. These videos served as the first viewing material for the participants in both the high- and low- difficulty conditions.** The combinations of video files presented in each condition are detailed in 'Stimulus.xlsx.'
(ZIP)

**S2 Video. In the low-difficulty condition, these videos were presented second to the participants, designed to facilitate easier estimation of the true position of the target card.**
(ZIP)

**S3 Video. These videos were presented third to the participants in both the high- and low-difficulty conditions.** These were designed to reveal hints of the card magic technique from 9 seconds after the beginning of the video until the end.
(ZIP)

## Author contributions

**Conceptualization:** Kazutaka Ueda, Xiaoxiang Wu, Hideyoshi Yanagisawa.

**Funding acquisition:** Hideyoshi Yanagisawa.

**Investigation:** Kazutaka Ueda, Xiaoxiang Wu.

**Methodology:** Kazutaka Ueda, Xiaoxiang Wu, Hideyoshi Yanagisawa.

**Project administration:** Hideyoshi Yanagisawa.

**Supervision:** Hideyoshi Yanagisawa.

**Visualization:** Kazutaka Ueda, Xiaoxiang Wu, Hideyoshi Yanagisawa.

**Writing – original draft:** Kazutaka Ueda, Xiaoxiang Wu, Hideyoshi Yanagisawa.

**Writing – review & editing:** Kazutaka Ueda, Hideyoshi Yanagisawa.

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
