## [Decision Letter · Decision Letter 0]

PONE-D-24-33040Neural mechanisms of emotion during shifting perspectives and recognizing new information: An fMRI studyPLOS ONE

Dear Dr. Yanagisawa,

Thank you for submitting your manuscript to PLOS ONE. After careful consideration, we feel that it has merit but does not fully meet PLOS ONE’s publication criteria as it currently stands. Therefore, we invite you to submit a revised version of the manuscript that addresses the points raised during the review process.

We look forward to receiving your revised manuscript.

Kind regards,

Wi Hoon Jung, PhD

Academic Editor

PLOS ONE

Journal Requirements:

“This study was supported by KAKEN grant number 21H03528 from the Japan Society for the Promotion of Science”

“Initials of author who received the grant: Hideyoshi Yanagisawa (H.Y)

Grant number: 21H03528

Full name of funder: Japan Society for the Promotion of Science

URL of the funder website: https://www.jsps.go.jp/english/e-grants/ ”

Additional Editor Comments (if provided):

Please review the comments made by the reviewer below regarding this study and revise the manuscript. In particular, we ask that you revise the conclusions and interpretations of the results of this study.

Reviewers' comments:

Reviewer's Responses to Questions

**Comments to the Author**

1. Is the manuscript technically sound, and do the data support the conclusions?

Reviewer #1: Partly

2. Has the statistical analysis been performed appropriately and rigorously? 

Reviewer #1: Yes

3. Have the authors made all data underlying the findings in their manuscript fully available?

Reviewer #1: Yes

4. Is the manuscript presented in an intelligible fashion and written in standard English?

Reviewer #1: Yes

5. Review Comments to the Author

Reviewer #1: The paper "Neural mechanisms of emotion during shifting perspectives and recognizing new information: An fMRI study" applies the free energy principle and utilizes fMRI to investigate a mathematical model's ability to predict emotions when individuals shift perspectives to acknowledge alternative information about an event. The study concludes that interest relates to unexpected yet easily comprehensible new concepts.

The fMRI study demonstrates a well-structured database design, and the conclusions are statistically supported. However, some issues require revision. Once these concerns are addressed, I recommend the manuscript for publication.

The authors assert (lines 352-354) that "the amygdala [21–25] and hippocampus, which showed increased connectivity in the high-difficulty condition, are related to positive emotion, and these results are consistent with the model predictions." However, literature does not support this conclusion. While the amygdala can respond to positive stimuli, it is preferentially activates in fearful conditions. Therefore, a more accurate conclusion would be a generalized response to emotionally valenced stimuli.

The authors need to justify their conclusion (lines 349-350) as to why shifting from recognition based on certain beliefs to recognition based on uncertain beliefs results in a smaller recognition distance and easier recognition. In reality, the expected response is a smaller recognition distance for recognition based on certain beliefs.

Creativity is not uniformly distributed among individuals in society. Therefore, providing information on the study subjects' backgrounds (such as their familiarity with card tricks or their occupations in creative fields) could enhance the applicability and conclusions of the results.

While the data presented support the hypothesis, the relationship is not straightforward. Exploring alternative explanations for the results would create a stronger argument.

6. PLOS authors have the option to publish the peer review history of their article (what does this mean? ). If published, this will include your full peer review and any attached files.

**Do you want your identity to be public for this peer review?** For information about this choice, including consent withdrawal, please see our Privacy Policy .

Reviewer #1: No

---

## [Author Response · Author response to Decision Letter 1]

23 May 2025

Manuscript ID: PONE-D-24-33040

Title: Neural mechanisms of emotion during shifting perspectives and recognizing new information: An fMRI study

Dear Editor,

We are most grateful to you and the reviewers for the helpful comments on the original version of our manuscript. In response to the reviewers' comments, we have revised the manuscript.

We hope that the revised version of our paper is now suitable for publication in PLOS One and we look forward to hearing from you at your earliest convenience.

Sincerely,

Hideyoshi Yanagisawa, Ph.D.

Our responses to the Editor and reviewers have been uploaded respectively.

Reviewer #1 Comments and Responses

We are grateful to reviewer #1 for the critical comments and useful suggestions that have helped us to improve our paper considerably. As indicated in the responses that follow, we have taken the comments and suggestions into account in the revised version of our paper.

Reviewer #1’s Comment 1

The authors assert (lines 352-354) that "the amygdala [21–25] and hippocampus, which showed increased connectivity in the high-difficulty condition, are related to positive emotion, and these results are consistent with the model predictions." However, literature does not support this conclusion. While the amygdala can respond to positive stimuli, it is preferentially activates in fearful conditions. Therefore, a more accurate conclusion would be a generalized response to emotionally valenced stimuli.

Response 1

We sincerely thank the reviewer for identifying this important issue.

Upon reviewing our manuscript, we found that the experimental result itself was correctly analyzed and presented in the figures and results section, but the description in the Discussion (original manuscript, lines 352–354) mistakenly stated that the increased connectivity between the amygdala and hippocampus occurred in the high-difficulty condition. In fact, as correctly shown in Figure 8b and described in the results section, this increased connectivity was observed in the low-difficulty condition, in which participants were able to more easily reinterpret the card trick from a different perspective. We have now corrected the text in the Discussion to accurately reflect this.

Furthermore, we appreciate the reviewer’s clarification regarding the role of the amygdala in emotion processing. We have revised the interpretation to avoid implying a specific emotional valence, as the amygdala is known to respond to both positive and negative emotional stimuli. The revised wording more accurately reflects the scope of the empirical findings and their relationship to our theoretical model, emphasizing emotional salience and cognitive engagement rather than valence-specific emotion.

We revised the sentence in the Discussion section (lines 353–359) to read:

“Additionally, in the low-difficulty condition, increased functional connectivity between the amygdala [21–25] and hippocampus was observed. The amygdala is involved in processing emotionally salient stimuli, regardless of valence. This result may reflect heightened emotional salience during successful perspective shifts. This interpretation aligns with the model’s assumption that when perspective is shifted from a certain belief to a more uncertain belief, and the recognition distance is relatively small, greater disfluency reduction and emotional engagement are more likely to occur.”

Reviewer #1’s Comment 2

The authors need to justify their conclusion (lines 349-350) as to why shifting from recognition based on certain beliefs to recognition based on uncertain beliefs results in a smaller recognition distance and easier recognition. In reality, the expected response is a smaller recognition distance for recognition based on certain beliefs.

Response 2

Thank you for raising this important point. We agree that the original wording in lines 349–350 in original manuscript may have inadvertently suggested a direct causal relationship—that shifting from certain to uncertain beliefs inherently leads to smaller recognition distances and easier recognition.

In fact, our model assumes a more nuanced prediction: when a recognition shift occurs from a certain to an uncertain belief, and the recognition distance is small, then disfluency reduction is greater, leading to more cognitively fluent recognition and increased emotional engagement.

We also wish to clarify that this prediction was explicitly stated in the Introduction of the original manuscript. On lines 134–137, we wrote:

“Specifically, by measuring human brain functions, we tested the prediction that a smaller recognition distance (i.e., easier recognition) would produce more positive emotions, such as interest, when shifting from recognition based on certain beliefs to recognition based on uncertain beliefs.”

To improve clarity and more accurately reflect the theoretical framework, we have revised the Discussion section (lines 346–353) to the following:

“These results suggest that when recognition shifts from a certain belief to a more uncertain one, greater activity in the cingulate gyrus [18–20] and hippocampus [26] is observed, particularly in cases in which the recognition distance is relatively small and recognition becomes easier. Given that both regions have been previously associated with the processing of positive emotional experiences, this neural pattern supports our model’s prediction that recognition distance, rather than belief uncertainty alone, is a key determinant of disfluency reduction and emotionally engaging responses.”

This revision eliminates the oversimplified causality in the original sentence and more accurately conveys the model’s condition-dependent prediction.

Reviewer #1’s Comment 3

Creativity is not uniformly distributed among individuals in society. Therefore, providing information on the study subjects' backgrounds (such as their familiarity with card tricks or their occupations in creative fields) could enhance the applicability and conclusions of the results.

Response 3

Thank you for this valuable suggestion. As described in the Participants section of the Methods, all participants were individuals who had not mastered card magic skills and thus had no specialized expertise in the specific domain of the task. However, we did not assess the participants’ broader creative profiles, including their occupations in creative fields or general familiarity with visual illusions or perspective-shifting tasks. We have now acknowledged this as a limitation in the Discussion section (lines 371-376):

“A limitation of this experiment is that we did not collect information about participants’ prior experience with magic tricks, their occupations in creative fields, or their general creative background. Although all participants had not mastered card magic skills, individual differences in creative training, profession, or familiarity with similar illusions could influence the ease of recognition or emotional responses. Future studies should consider assessing such background factors to better account for variance in emotional and cognitive responses.”

Reviewer #1’s Comment 4

While the data presented support the hypothesis, the relationship is not straightforward. Exploring alternative explanations for the results would create a stronger argument.

Response 4

Thank you for this thoughtful and constructive comment. We agree that the relationship between the observed brain activity and our model’s predictions may not be entirely straightforward and that alternative explanations merit consideration.

In the revised manuscript, we have added a paragraph in the Discussion (lines 360-370) acknowledging this and outlining several alternative interpretations.

“Although our interpretation emphasizes that increased activation and connectivity in regions such as the anterior cingulate cortex, hippocampus, and amygdala may reflect disfluency reduction and emotional engagement during successful perspective shifts, we acknowledge that other interpretations are also plausible. For example, anterior cingulate cortex activity may reflect general cognitive control or conflict monitoring [28] in response to updating beliefs. Hippocampal activity may reflect indicate episodic memory retrieval [27] or novelty processing [29]. Similarly, increased connectivity between the amygdala and hippocampus in the low-difficulty condition may reflect enhanced attentional allocation to emotionally meaningful content [30]. These possibilities suggest that the observed neural responses may involve a combination of emotional, mnemonic, and attentional mechanisms. Future studies could incorporate more targeted designs to disentangle these processes.”

References

18. Lane RD, Fink GR, Chau PM-L, Dolan RJ. Neural activation during selective attention to subjective emotional responses. NeuroReport. 1997;8: 3969–3972. doi: 10.1097/00001756-199712220-00024, Pubmed:9462476.

19. Matsunaga M, Kawamichi H, Koike T, Yoshihara K, Yoshida Y, Takahashi HK, et al. Structural and functional associations of the rostral anterior cingulate cortex with subjective happiness. Neuroimage. 2016;134: 132–141. doi: 10.1016/j.neuroimage.2016.04.020, Pubmed:27085503.

20. Scharnowski F, Nicholson AA, Pichon S, Rosa MJ, Rey G, Eickhoff SB, et al. The role of the subgenual anterior cingulate cortex in dorsomedial prefrontal–amygdala neural circuitry during positive‐social emotion regulation. Hum Brain Mapp. 2020;41: 3100–3118. doi: 10.1002/hbm.25001, Pubmed:32309893.

21. Baxter MG, Murray EA. The amygdala and reward. Nat Rev Neurosci. 2002;3: 563–573. doi: 10.1038/nrn875, Pubmed:12094212.

22. Bonnet L, Comte A, Tatu L, Millot JL, Moulin T, Medeiros de Bustos E. The role of the amygdala in the perception of positive emotions: an “intensity detector”. Front Behav Neurosci. 2015;9: 178. doi: 10.3389/fnbeh.2015.00178, Pubmed:26217205.

23. Canli T, Sivers H, Whitfield SL, Gotlib IH, Gabrieli JDE. Amygdala response to happy faces as a function of extraversion. Science. 2002;296: 2191. doi: 10.1126/science.1068749, Pubmed:12077407.

24. Koepp MJ, Hammers A, Lawrence AD, Asselin MC, Grasby PM, Bench CJ. Evidence for endogenous opioid release in the amygdala during positive emotion. Neuroimage. 2009;44: 252–256. doi: 10.1016/j.neuroimage.2008.08.032, Pubmed:18809501.

25. Murray EA. The amygdala, reward and emotion. Trends Cogn Sci. 2007;11: 489–497. doi: 10.1016/j.tics.2007.08.013, Pubmed:17988930.

26. Gruber MJ, Gelman BD, Ranganath C. States of curiosity modulate hippocampus-dependent learning via the dopaminergic circuit. Neuron. 2014;84: 486–496. doi: 10.1016/j.neuron.2014.08.060, Pubmed:25284006.

28. Kerns JG, Cohen JD, MacDonald III AW, Cho RY, Stenger VA, Carter CS. Anterior cingulate conflict monitoring and adjustments in control. Science. 2004;303: 1023–1026. doi: 10.1126/science.1089910

29. Duncan K, Ketz N, Inati SJ, Davachi L. Evidence for area CA1 as a match/mismatch detector: A high‐resolution fMRI study of the human hippocampus. Hippocampus. 2012;22: 389–398. doi: 10.1002/hipo.20933

30. Zheng J, Anderson KL, Leal SL, Shestyuk A, Gulsen G, Mnatsakanyan L, Vadera S, Hsu FPK, Yassa MA, Knight RT. Lin JJ. Amygdala-hippocampal dynamics during salient information processing. Nat Commun. 2017;8: 14413. doi: 10.1038/ncomms14413

---

## [Decision Letter · Decision Letter 1]

Neural mechanisms of emotion during shifting perspectives and recognizing new information: An fMRI study

PONE-D-24-33040R1

Dear Dr. Yanagisawa,

We’re pleased to inform you that your manuscript has been judged scientifically suitable for publication and will be formally accepted for publication once it meets all outstanding technical requirements.

Kind regards,

Wi Hoon Jung, PhD

Academic Editor

PLOS ONE

Additional Editor Comments (optional):

Reviewers' comments:

Reviewer's Responses to Questions

**Comments to the Author**

1. If the authors have adequately addressed your comments raised in a previous round of review and you feel that this manuscript is now acceptable for publication, you may indicate that here to bypass the “Comments to the Author” section, enter your conflict of interest statement in the “Confidential to Editor” section, and submit your "Accept" recommendation.

Reviewer #1: All comments have been addressed

2. Is the manuscript technically sound, and do the data support the conclusions?

Reviewer #1: Yes

3. Has the statistical analysis been performed appropriately and rigorously? 

Reviewer #1: Yes

4. Have the authors made all data underlying the findings in their manuscript fully available?

Reviewer #1: Yes

5. Is the manuscript presented in an intelligible fashion and written in standard English?

Reviewer #1: Yes

6. Review Comments to the Author

Reviewer #1: Thank you for your thoughtful responses to my comments on the manuscript. The revisions effectively address the concerns I raised, and they have significantly strengthened the manuscript. With these improvements, I am pleased to recommend the manuscript for publication.

7. PLOS authors have the option to publish the peer review history of their article (what does this mean? ). If published, this will include your full peer review and any attached files.

**Do you want your identity to be public for this peer review?** For information about this choice, including consent withdrawal, please see our Privacy Policy .

Reviewer #1: **Yes: ** Eva K Deli

---

## [Editor Report · Acceptance letter]

PONE-D-24-33040R1

PLOS ONE

Dear Dr. Yanagisawa,

I'm pleased to inform you that your manuscript has been deemed suitable for publication in PLOS ONE. Congratulations! Your manuscript is now being handed over to our production team.

Kind regards,

on behalf of

Dr. Wi Hoon Jung

Academic Editor

PLOS ONE